# Application of Life Cycle Assessment in the Environmental Study of Sustainable Ceramic Bricks Made with 'alperujo' (Olive Pomace)

**Ana B. López-García** [1] , **Teresa Cotes-Palomino** [1] , **Manuel Uceda-Rodríguez** [1] , **José Manuel Moreno-Maroto** [1] , **Carlos Javier Cobo-Ceacero** [1] , **N. M. Fernanda Andreola** [2] and **Carmen Martínez-García** [1,*]

[1] Department of Chemical, Environmental and Materials Engineering, Higher Polytechnic School of Linares, Scientific and Technological Campus of Linares, University of Jaen, 23700 Linares (Jaén), Spain; ablopez@ujaen.es (A.B.L.-G.); mtcotes@ujaen.es (T.C.-P.); muceda@ujaen.es (M.U.-R.); jmmaroto@ujaen.es (J.M.M.-M.); cjcobo@ujaen.es (C.J.C.-C.)

[2] Department of Engineering 'Enzo Ferrari', University of Modena and Reggio Emilia, Via Pietro Vivarelli 10, 41125 Modena, Italy; fernandanora.andreola@unimore.it

\* Correspondence: cmartin@ujaen.es; Tel.: +34-953-648-548

**Abstract:** Investigations on the application of Life Cycle Assessment (LCA) to the construction sector have shown that the environmental impact of construction products can be significantly reduced. To achieve this, the use of best available techniques and eco-innovation in production plants must be promoted. In this way, the use of finite natural resources can be replaced by waste generated in other production processes, preferably available locally, stimulating the creation of more sustainable products. Conducting a comparative LCA study between the traditional ceramic brick manufacturing process and the ceramic brick manufacturing process incorporating 'alperujo' (waste generated in the virgin oil extraction process), is an inevitable step to achieve the integration of circularity and eco-innovation in the production system of traditional ceramic materials, through the CML(Centrum voor Milieukunde Leiden) and IPCC(The Intergovernmental Panel on Climate Change) methodology. The obtained results suggest that the environmental benefits in this practice are very limited, even taking into account the contribution of different amounts of this waste to the production of bricks.

**Keywords:** circular economy; waste recycling; Life Cycle Assessment; olive pomace; brick industry

## 1. Introduction

Human influence on atmospheric quality has been increasing in recent years. If left unchecked, through the use of rigorous mitigation activities, climate change will increase the likelihood of severe, widespread, and irreversible impacts on people and ecosystems [1]. To truly minimize the risks of climate change, substantial and long-term reductions in greenhouse gas emissions are needed. Obviously, the ecological transition that is being pursued will involve the transformation of the country's major economic sectors [2].

In Europe, the construction sector is responsible for 40% of $CO_2$ emissions, 30% of raw material consumption, 20% of water consumption, 30% of waste generation, and a significant part of land occupation. The need to transform the construction sector involves reducing emissions of polluting gases through the use of new materials with a low or zero carbon footprint and, in addition, promoting the progress of the circular economy by reusing and incorporating materials from waste. Therefore, it is more than evident the necessary transition from the current linear production system to a production system based on a circular economy that allows the search for new alternatives with the main objective of curbing the consumption of natural reserves and the increase of $CO_2$ emissions, as well as moving towards greater sustainability [3,4].

On the other hand, the olive pomace (the so called 'alperujo') can represent up to 80% of the olive production destined to olive oil mills for olive oil extraction [5]. According

to data corresponding to the seasons comprised in the period 2015–2018, the production of olive oil and table olives was 1.2 million and 550,000 tonnes in Spain, representing approximately 40% and 20% of the world total [6–8]. This means that an estimated 11 million tonnes of olive pomace can be produced worldwide.

In this sense, the main objective of the present work is to carry out a comparative study of the Life Cycle Assessment (LCA) of traditional ceramic materials with respect to sustainable ceramic materials manufactured with olive pomace, waste generated in the virgin oil extraction process, which allows us to determine the global impact of the life cycle of the different products evaluated. For this purpose, the LCA of traditionally manufactured bricks was compared with the LCA of bricks in which 3, 7, and 10% of clay has been replaced by alperujo (ALP), based on the definition of the objective, scope, limits, and functional unit of the analysis, the performance of the Life Cycle Inventory (LCI) analysis, the Life Cycle Impact Assessment (LCIA) and, finally, the interpretation of the results obtained. The purpose of this is to establish the best environmentally sustainable options, to increase the amount of information available on the product and the process, and to identify points for improvement that can be proposed in the future. Olive pomace has been used because the main output of this by-product is its use as fuel with the consequent generation of a large amount of ash as final residue (between 4 and 8% of the waste burned). The common disposal of this biomass ash in landfills located next to power plants is an environmentally unattractive alternative [9–12].

## 2. Methodology

LCA is the most widely used methodology for seeking environmental solutions, minimizing the carbon footprint, and avoiding the production of impacts derived from the manufacture of products or services. LCA makes it possible to optimize inputs (materials and energy) and minimize outputs (waste and environmental impacts) of the activity under study. LCA technique has great potential in the study of the environmental impacts associated with the production of new materials [13] or conventional materials with applications in different sectors [14]. This methodology is regulated by the ISO 14040 and ISO 14044 standards [15,16], and its main function is to establish a common basis from which to set inputs and outputs to the biosphere and technosphere through four steps for each approach: (i) Definition of the objective and scope; (ii) inventory analysis; (iii) impact assessment; and (iv) interpretation of the results. LCAs performed to the brick manufacturing industry have shown that the main environmental impacts occur due to energy consumption derived from the firing process [17,18], finding values for climate change of fired clay bricks ranging from 132 to 295 kg $CO_2$ equivalent/tonne of brick, oscillations that are mainly attributed to the scope of the LCA, the characteristics of the firing process, and the quality of the bricks. There are few studies based on the use of LCA to determine the environmental benefits of waste incorporation in fired bricks [19–23].

### 2.1. Definition of the Objective and Scope of the Study

The objective of the present work has been to provide information on the environmental consequences of the brick production system from cradle to gate by comparing bricks produced in the traditional way with bricks in which certain amounts of clay have been substituted by olive pomace (3, 7, and 10% by weight), for a reference time period of 100 years, measuring their impact using the IPCC and CML methodologies.

The cradle-to-gate approach analyzes the impact from the extraction of raw materials, the production of materials and product parts, and until the final product leaves the factory. The useful life and end-of-life stages are not the subject of study in this work.

In this Life Cycle Analysis, the industrial manufacturing of 1 kg of traditional brick was compared to 1 kg of brick incorporating 3, 7, and 10% by weight of olive pomace for a period of 100 years. Approximate dimensions of the samples were $117 \times 28 \times 17$ mm. The firing temperature defined in this analysis is 850 °C because the bricks fired at these temperatures presented the best technological and thermal properties in previously developed works [23].

Figures 1 and 2 show images of the specimens produced in the laboratory before and after the sintering process, at the firing temperature of 850 °C and with different percentages of olive pomace added (3, 7, and 10%).

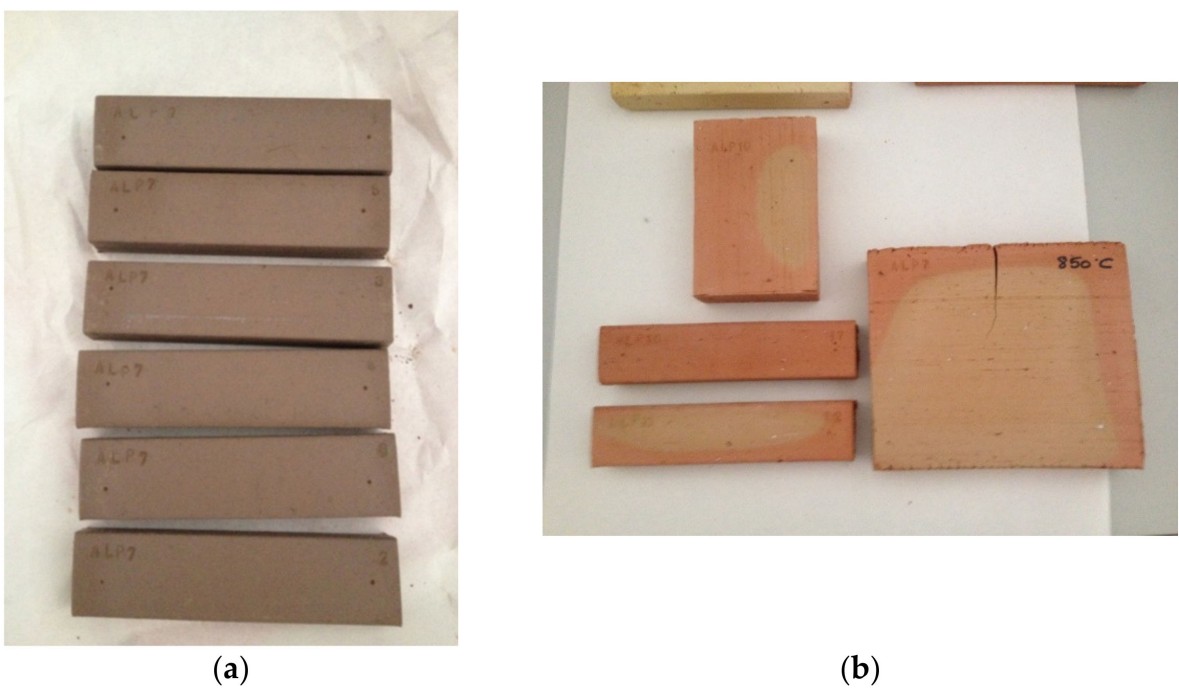

(a)                                                        (b)

**Figure 1.** Specimens before (**a**) and after (**b**) sintering.

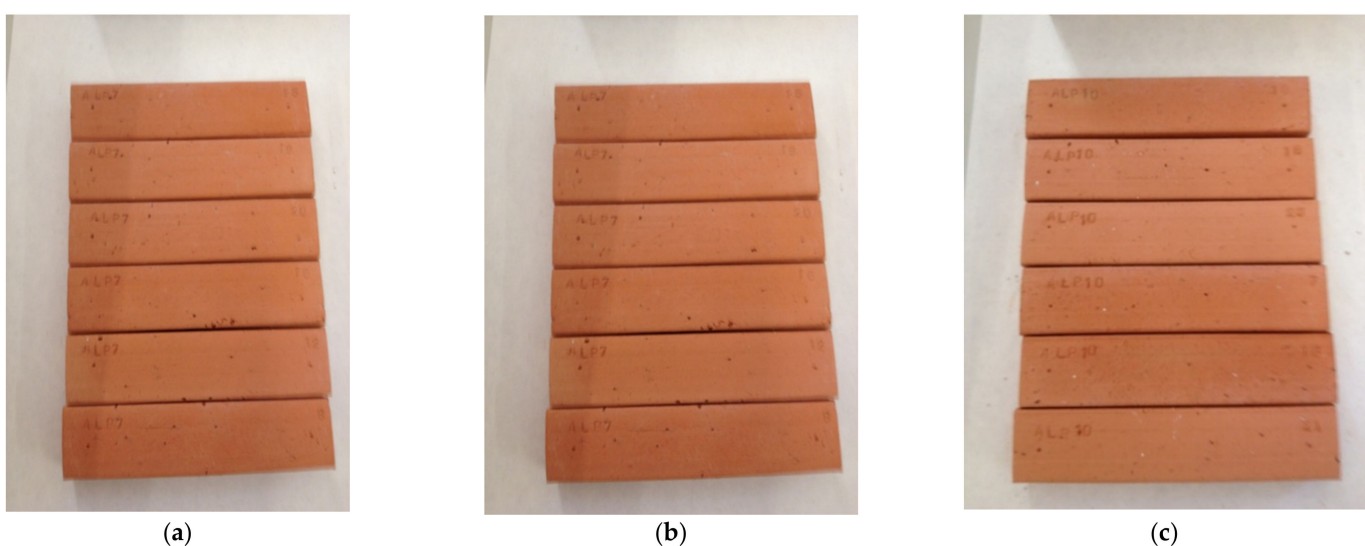

(a)                                        (b)                                        (c)

**Figure 2.** Specimens fired at 850 °C; (**a**) 3% ALP; (**b**) 7% ALP; (**c**)10% ALP.

### 2.2. Life Cycle Inventory

Table 1 shows the modelling performed in SIMAPRO for each of the processes involved in the manufacture of bricks by the traditional method and the manufacture of bricks with the addition of olive pomace, following the scheme represented in Figure 3, where the system limits taken as a reference in this study are indicated.

**Table 1.** Inventory data adjusted to 1 kg of bricks for the different phases analyzed: Raw materials; Extraction and transport; Manufacturing (energy and material inputs at bricks manufacturing plant) and Manufacturing (direct emissions associated with upstream processing).

| Elementary Flow | Units | 0% | 3% | 7% | 10% | LCIA Dataset |
|---|---|---|---|---|---|---|
| | | | *Raw materials* | | | |
| Clay | kg | 1.11 | 107.48 | 103.04 | 0.9972 | Clay {GLO} \| market for \| Alloc Def, U |
| Olive pomace | kg | - | 0.03324 | 0.07756 | 0.1108 | - |
| Water | m$^3$ | 0.0000736 | 0.0000736 | 0.0000736 | 0.0000736 | Water, well, in ground, ES |
| | kg | 0.0272 | 0.0272 | 0.0272 | 0.0272 | Tap water {GLO} \| market group for \| Alloc Def, U |
| Limestone, crushed, for mill | kg | 0.000396 | 0.000396 | 0.000396 | 0.000396 | Limestone, crushed, for mill {GLO} \| market for \| Alloc Def, U |
| Lime, packed | kg | 0.0239 | 0.0239 | 0.0239 | 0.0239 | Lime, packed {GLO} \| market for \| Alloc Def, U |
| Sand | kg | 0.0147 | 0.0147 | 0.0147 | 0.0147 | Sand {GLO} \| market for \| Alloc Def, U |
| | | | *Extraction and transport* | | | |
| Lubricating oil | kg | 0.0000132 | 0.0000132 | 0.0000132 | 0.0000132 | Lubricating oil {GLO} \| market for \| Alloc Def, U |
| Extraction plant | p | 0.0000000002 | 0.0000000002 | 0.0000000002 | 0.0000000002 | Clay pit infrastructure {GLO} \| market for \| Alloc Def, U |
| Natural gas | m$^3$ | 0.000047863 | 0.000047863 | 0.000047863 | 0.000047863 | Natural gas, high pressure {GLO} \| market group for \| Alloc Def, U |
| Electricity | kWh | 0.00056 | 0.00056 | 0.00056 | 0.00056 | Electricity, medium voltage {GLO} \| market group for \| Alloc Def, U |
| Light fuel oil | kg | 0.00541 | 0.00541 | 0.00541 | 0.00541 | Light fuel oil {RoW} \| market for \| Alloc Def, U |
| Transport raw materials | kgkm | 50 | 50 | 50 | 50 | Transport, freight, lorry 16–32 metric tonne, EURO5 {GLO} \| market for \| Alloc Def, U |
| Heavy fuel oil | kg | 0.000381 | 0.000381 | 0.000381 | 0.000381 | Heavy fuel oil {RoW} \| market for \| Alloc Def, U |
| | | | *Energy and material inputs at bricks manufacturing plant* | | | |
| Packaging film | kg | 0.000542 | 0.000542 | 0.000542 | 0.000542 | Packaging film, low density polyethylene {GLO} \| market for \| Alloc Def, U |
| Polyethylene | kg | 0.000000858 | 0.000000858 | 0.000000858 | 0.000000858 | Polyethylene, high density, granulate {GLO} \| market for \| Alloc Def, U |
| EUR-flat pallet | p | 0.0000161 | 0.0000161 | 0.0000161 | 0.0000161 | EUR-flat pallet {GLO} \| market for \| Alloc Def, U |
| Natural gas | m$^3$ | 0.047576 | 0.033674 | 0.015139 | 0.001237 | Natural gas, high pressure {GLO} \| market group for \| Alloc Def, U |

**Table 1.** *Cont.*

| Elementary Flow | Units | 0% | 3% | 7% | 10% | LCIA Dataset |
|---|---|---|---|---|---|---|
| Sheet rolling | kg | 0.000000157 | 0.000000157 | 0.000000157 | 0.000000157 | Sheet rolling, chromium steel {GLO} \| market for \| Alloc Def, U |
| Steel, low-alloyed, hot rolled | kg | 0.0000306 | 0.0000306 | 0.0000306 | 0.0000306 | Steel, low-alloyed, hot rolled {GLO} \| market for \| Alloc Def, U |
| Sheet rolling, steel | kg | 0.0000157 | 0.0000157 | 0.0000157 | 0.0000157 | Sheet rolling, steel {GLO} \| market for \| Alloc Def, U |
| Polystyrene, expandable | kg | 0.000352 | 0.000352 | 0.000352 | 0.000352 | Polystyrene, expandable {GLO} \| market for \| Alloc Def, U |
| Electricity | kWh | 0.047971 | 0.033954 | 0.015265 | 0.001248 | Electricity, medium voltage {GLO} \| market group for \| Alloc Def, U |
| *Direct emissions associated with upstream processing* | | | | | | |
| Nitrogen oxides | kg | 0.00026 | 0.00026 | 0.00026 | 0.00026 | Emissions to air—Nitrogen oxides |
| Benzene | kg | 0.00000296 | 0.00000296 | 0.00000296 | 0.00000296 | Emissions to air—Benzene |
| Sulfur dioxide | kg | 0.0000998 | 0.0000998 | 0.0000998 | 0.0000998 | Emissions to air—Sulfur dioxide |
| Carbon monoxide, fossil | kg | 0.000391 | 0.000391 | 0.000391 | 0.000391 | Emissions to air—Carbon monoxide, fossil |
| Particulates | kg | 0.000014 | 0.000014 | 0.000014 | 0.000014 | Emissions to air—Particulates, <2.5 um |
|  | kg | 0.00000468 | 0.00000468 | 0.00000468 | 0.00000468 | Emissions to air—Particulates, >10 um |
| Formaldehyde | kg | 0.0000164 | 0.0000164 | 0.0000164 | 0.0000164 | Emissions to air—Formaldehyde |
| Hydrogen fluoride | kg | 0.0000106 | 0.0000106 | 0.0000106 | 0.0000106 | Emissions to air—Hydrogen fluoride |
| Water | $m^3$ | 0.00001512 | 0.00001512 | 0.00001512 | 0.00001512 | Emissions to air—Water/$m^3$ |
| Hydrogen chloride | kg | 0.0000122 | 0.0000122 | 0.0000122 | 0.0000122 | Emissions to air—Hydrogen chloride |
| NMVOC | kg | 0.0000763 | 0.0000763 | 0.0000763 | 0.0000763 | Emissions to air—NMVOC, non-methane volatile organic compounds, unspecified |
| Carbon dioxide, fossil | kg | 0.18 | 0.2336 | 0.3051 | 0.3588 | Emissions to air—Carbon dioxide, fossil |
| Phenol | kg | 0.00000013 | 0.00000013 | 0.00000013 | 0.00000013 | Emissions to air—Phenol |
| Water | $m^3$ | 0.00008568 | 0.00008568 | 0.00008568 | 0.00008568 | Emissions to water—Water, RoW |

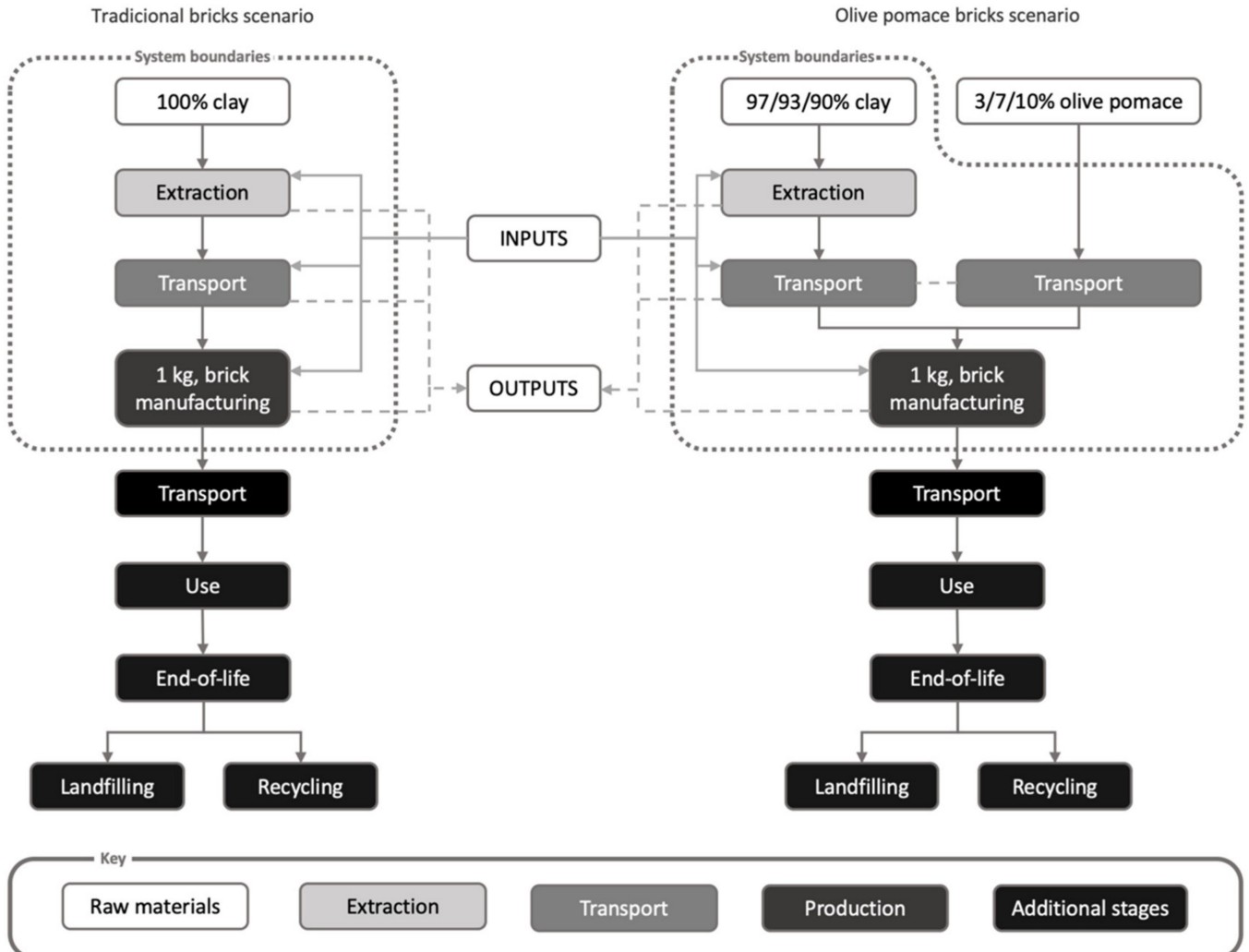

**Figure 3.** Life cycle flow chart of traditional bricks and bricks incorporating olive pomace in the different scenarios considered.

Figure 3 illustrates in schematic form the production process of traditional bricks on the left and sustainable bricks using waste on the right. In this study, the system boundaries have been considered from raw material extraction to brick production. During the 'Extraction', 'Transport', and 'Manufacture' phases, a series of inputs and outputs of material and energy will occur, which are illustrated in the diagram and which, to a greater or lesser extent, will be shared by both processes. The differential stage will be the 'Manufacture' stage, where the exothermic power of the olive pomace used allows the energy cost to be reduced during the sintering process.

Table 1 illustrates the inventory data for the 'Raw materials' phase. Table 1 lists the impacts associated with the extraction of natural clay, which are those related to the quarry operation, while the impacts attributable to the extraction of olive pomace are assigned to the main product of the olive oil extraction process and are not considered due to the residual nature of this material. The same table shows the inventory data for the 'Transport' phase of the raw materials to the mill. Emissions from the transport of the residue have been assimilated as the corresponding part of the clay it replaces.

'Manufacturing' in Table 1 would be the last stage considered in this study. The data related to the manufacture of bricks in the factory by the traditional process as well as with bricks containing 3, 7, and 10% of olive pomace. It considers both energy consumption in the form of electricity, diesel, and natural gas in the brick manufacturing plant. Finally, Table 1 presents direct emissions in the form of $CO_2$, $NO_x$, $SO_2$, HF, and HCl generated by the thermal transformation of the raw materials (natural clay and waste residue) during the firing process.

*2.3. Impact Assessment Methodology*

As a support tool for the analysis and quantification of impacts, the SIMAPRO 8.3.0.0.0 program of the company PRé Consultants was used. The ECOINVENT database was used in its version 3 [24], and two evaluation methodologies were used. The first methodology applied was IPCC, which characterizes emissions according to their global warming potential, through the valuation of greenhouse gases, including carbon dioxide, methane, nitrogen oxides, and chlorofluorocarbons, among others. The potential is evaluated in terms of $CO_2$ eq, so that the emission of 1 kg of a particular greenhouse gas is expressed as the emission of 1 kg of carbon dioxide equivalent using established conversion factors. The second methodology used was CML 2000 V2.05, which analyzes abiotic resource depletion, acidification, eutrophication, global warming potential, ozone depletion, human toxicity, aquatic and terrestrial ecotoxicity, and photochemical oxidation as impact categories. The regional validity of the impact categories of the CML methodology is global, except for acidification and photochemical oxidation, which are based on European average values [25–27]. With respect to the latter methodology, the results have also been evaluated from a standardized point of view with respect to the environmental effects caused by an average European in one year [28].

## 3. Results and Discussion

Figure 4 and Table 2 illustrate the impacts obtained by IPCC-WGP-100 years for 1 kg of standard bricks produced from 100% natural clay and 1 kg of bricks produced with a 3, 7, and 10% by weight addition of olive pomace. The impact of the traditional system on global warming potential represents 260 kg $CO_2$ equivalent/tonne of bricks, a value comparable to those indicated by other authors such as [21–24], showing values of 221, 271, and 195 kg $CO_2$ equivalent/tonne of bricks, respectively. However, bricks with olive pomace have been shown to be environmentally less sustainable, showing higher climate change impact values than the traditional system. The increase in climate change potential of bricks with 3% compared to bricks obtained by the traditional system is 19.2% (310 kg $CO_2$ equivalent/tonne of bricks), increasing to 44.2% and 63% for bricks with 7 and 10% of olive pomace, respectively. The bricks with 3% of olive pomace have proven to be more environmentally sustainable than the rest of the bricks tested with the addition of wastewater residue for the established time period of 100 years.

Figure 5 shows the results obtained using the CML 2 baseline 2000 v2.05 methodology for a wider range of impact categories for traditional bricks. Figure 6 shows the impacts generated for the rest of the scenarios analyzed in this work in each of the ten selected environmental categories. Most of the impact, in all the scenarios tested, is attributable to the manufacturing phase, in particular the combustion of fossil fuels and the use of electricity during the firing process. The data on which these figures are based are shown in Table 3.

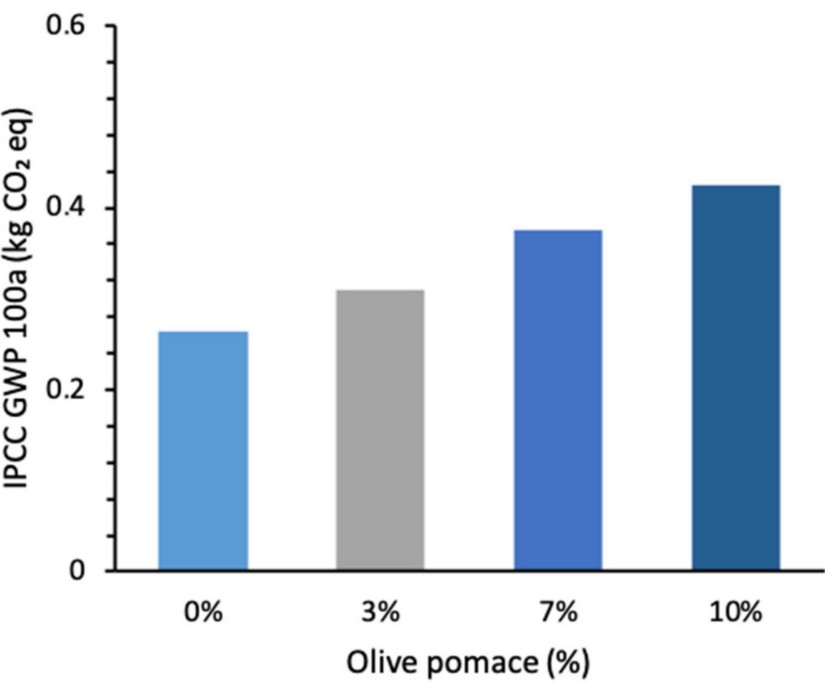

**Figure 4.** Climate change potential, 100 years (kg CO$_2$ eq).

**Table 2.** Climate change potential, 100 years (kg CO$_2$ eq).

| Olive Pomace (%) | IPCC GWP 100 a (kg CO$_2$ eq) |
|---|---|
| 0 | 0.263 |
| 3 | 0.310 |
| 7 | 0.375 |
| 10 | 0.424 |

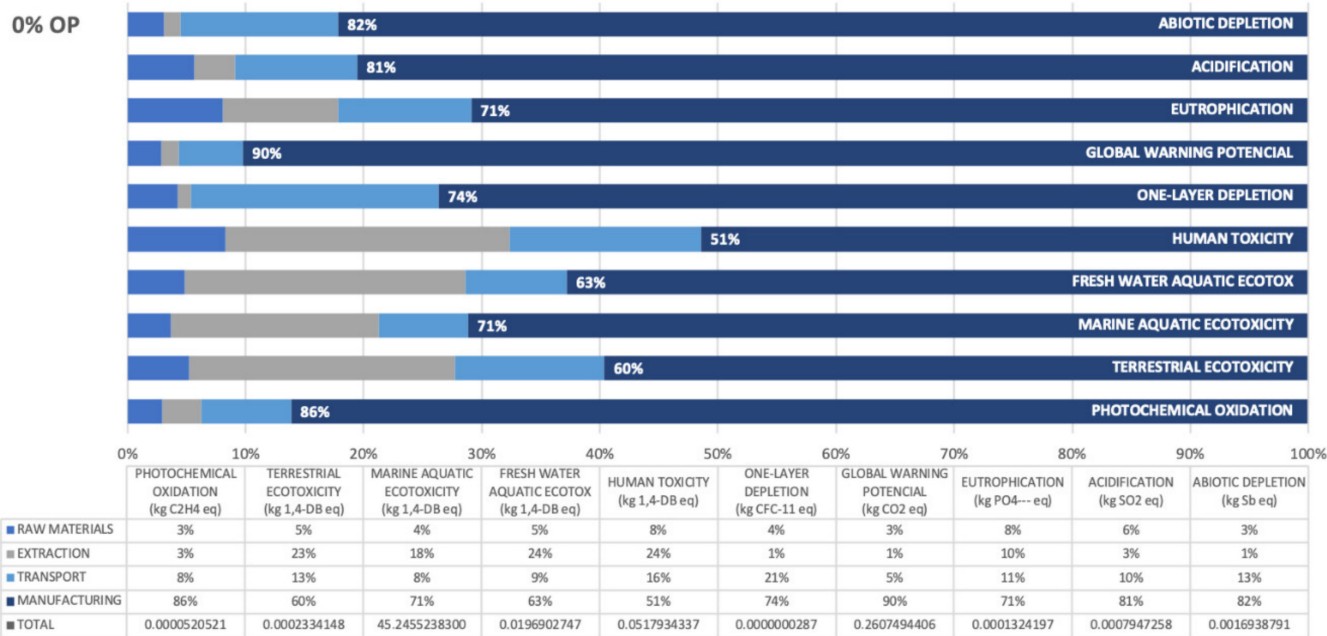

**Figure 5.** Impact distribution for each of the stages analyzed in characterization in traditional bricks.

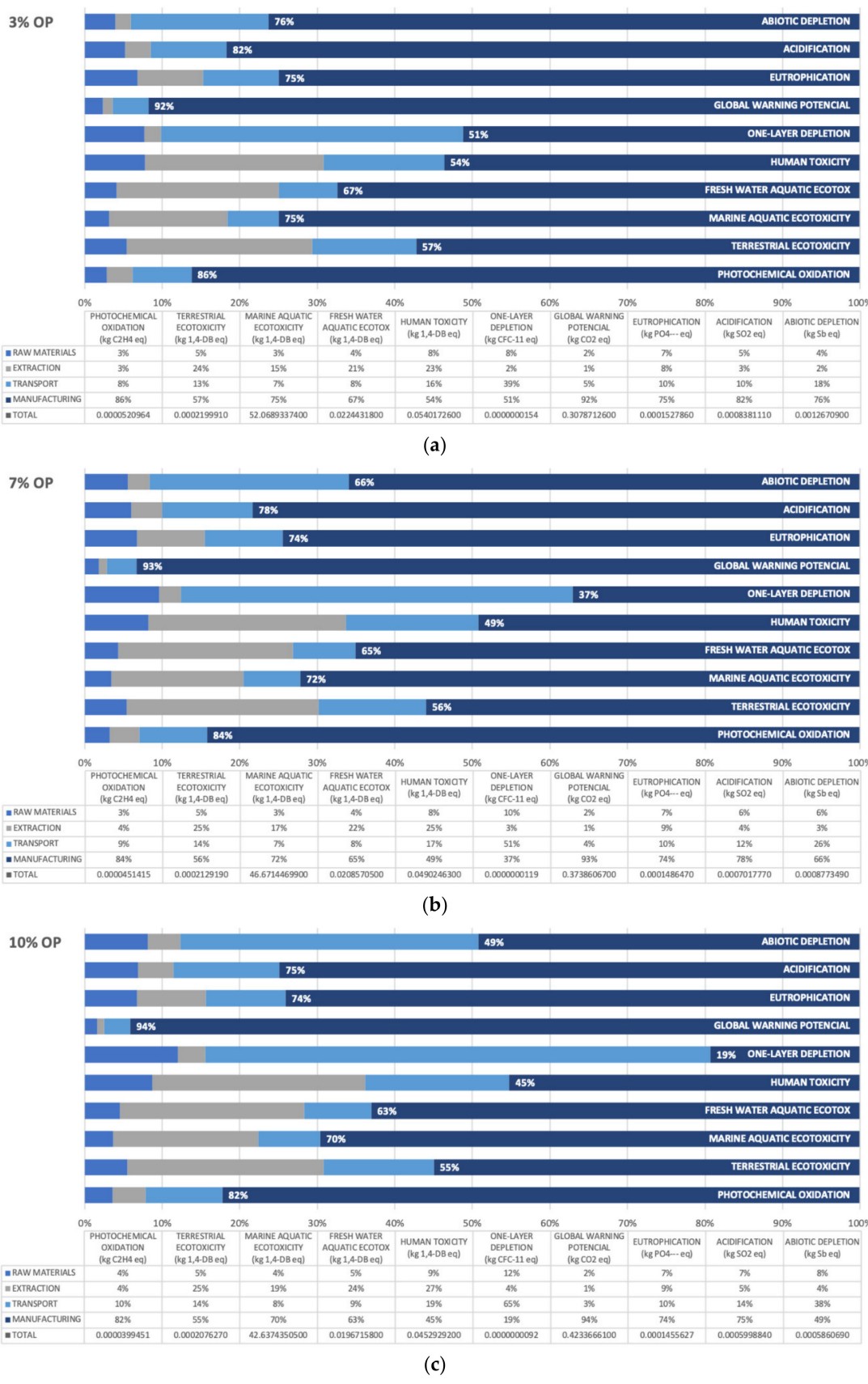

**Figure 6.** Impact distribution for each of the stages analyzed in characterization for bricks with olive pomace (**a**) bricks with 3% ALP; (**b**) bricks with 7% ALP; and (**c**) bricks with 10% ALP.

**Table 3.** Characterized impacts associated with 1 kg of bricks incorporating 0, 3, 7, and 10 wt% OP.

| Impact Categories | Units | Raw Materials | | | | Extraction | | | | Transport | | | | Manufacturing | | | |
|---|---|---|---|---|---|---|---|---|---|---|---|---|---|---|---|---|---|
| | | 0% | 3% | 7% | 10% | 0% | 3% | 7% | 10% | 0% | 3% | 7% | 10% | 0% | 3% | 7% | 10% |
| Abiotic Depletion | kg Sb eq | $5.20 \times 10^{-5}$ | $5.07 \times 10^{-5}$ | $4.90 \times 10^{-5}$ | $4.77 \times 10^{-5}$ | $2.45 \times 10^{-5}$ | $2.45 \times 10^{-5}$ | $2.45 \times 10^{-5}$ | $2.45 \times 10^{-5}$ | $2.25 \times 10^{-4}$ | $2.25 \times 10^{-4}$ | $2.25 \times 10^{-4}$ | $2.25 \times 10^{-4}$ | $1.39 \times 10^{-3}$ | $9.67 \times 10^{-4}$ | $5.79 \times 10^{-4}$ | $2.89 \times 10^{-4}$ |
| Acidification | kg $SO_2$ eq | $4.51 \times 10^{-5}$ | $4.40 \times 10^{-5}$ | $4.25 \times 10^{-5}$ | $4.14 \times 10^{-5}$ | $2.73 \times 10^{-5}$ | $2.73 \times 10^{-5}$ | $2.73 \times 10^{-5}$ | $2.73 \times 10^{-5}$ | $8.20 \times 10^{-5}$ | $8.20 \times 10^{-5}$ | $8.20 \times 10^{-5}$ | $8.20 \times 10^{-5}$ | $6.40 \times 10^{-4}$ | $6.85 \times 10^{-4}$ | $5.50 \times 10^{-4}$ | $4.49 \times 10^{-4}$ |
| Eutrophication | kg $PO_4^{3-}$ eq | $1.07 \times 10^{-5}$ | $1.04 \times 10^{-5}$ | $1.01 \times 10^{-5}$ | $9.85 \times 10^{-6}$ | $1.29 \times 10^{-5}$ | $1.29 \times 10^{-5}$ | $1.29 \times 10^{-5}$ | $1.29 \times 10^{-5}$ | $1.49 \times 10^{-5}$ | $1.49 \times 10^{-5}$ | $1.49 \times 10^{-5}$ | $1.49 \times 10^{-5}$ | $9.39 \times 10^{-5}$ | $1.15 \times 10^{-4}$ | $1.11 \times 10^{-4}$ | $1.08 \times 10^{-4}$ |
| Global Warning Potential | kg $CO_2$ eq | $7.41 \times 10^{-3}$ | $7.23 \times 10^{-3}$ | $6.99 \times 10^{-3}$ | $6.81 \times 10^{-3}$ | $3.82 \times 10^{-3}$ | $3.82 \times 10^{-3}$ | $3.82 \times 10^{-3}$ | $3.82 \times 10^{-3}$ | $1.43 \times 10^{-2}$ | $1.43 \times 10^{-2}$ | $1.43 \times 10^{-2}$ | $1.43 \times 10^{-2}$ | $2.35 \times 10^{-1}$ | $2.83 \times 10^{-1}$ | $3.49 \times 10^{-1}$ | $3.98 \times 10^{-1}$ |
| One-Layer Depletion | kg $CFC^{-11}$ eq | $1.22 \times 10^{-9}$ | $1.19 \times 10^{-9}$ | $1.15 \times 10^{-9}$ | $1.11 \times 10^{-9}$ | $3.26 \times 10^{-10}$ | $3.26 \times 10^{-10}$ | $3.26 \times 10^{-10}$ | $3.26 \times 10^{-10}$ | $6.01 \times 10^{-9}$ | $6.01 \times 10^{-9}$ | $6.01 \times 10^{-9}$ | $6.01 \times 10^{-9}$ | $2.11 \times 10^{-8}$ | $7.91 \times 10^{-9}$ | $4.40 \times 10^{-9}$ | $1.78 \times 10^{-9}$ |
| Human Toxicity | kg 1.4-DB eq | $4.30 \times 10^{-3}$ | $4.19 \times 10^{-3}$ | $4.05 \times 10^{-3}$ | $3.94 \times 10^{-3}$ | $1.24 \times 10^{-2}$ | $1.24 \times 10^{-2}$ | $1.24 \times 10^{-2}$ | $1.24 \times 10^{-2}$ | $8.41 \times 10^{-3}$ | $8.41 \times 10^{-3}$ | $8.41 \times 10^{-3}$ | $8.41 \times 10^{-3}$ | $2.66 \times 10^{-2}$ | $2.90 \times 10^{-2}$ | $2.41 \times 10^{-2}$ | $2.05 \times 10^{-2}$ |
| Fresh Water Aquatic Ecotoxicity | kg 1.4-DB eq | $9.51 \times 10^{-4}$ | $9.33 \times 10^{-4}$ | $9.10 \times 10^{-4}$ | $8.92 \times 10^{-4}$ | $4.68 \times 10^{-3}$ | $4.68 \times 10^{-3}$ | $4.68 \times 10^{-3}$ | $4.68 \times 10^{-3}$ | $1.69 \times 10^{-3}$ | $1.69 \times 10^{-3}$ | $1.69 \times 10^{-3}$ | $1.69 \times 10^{-3}$ | $1.24 \times 10^{-2}$ | $1.51 \times 10^{-2}$ | $1.36 \times 10^{-2}$ | $1.24 \times 10^{-2}$ |
| Marine Aquatic Ecotoxicity | kg 1.4-DB eq | 1.68 | 1.65 | 1.61 | 1.58 | 7.95 | 7.95 | 7.95 | 7.95 | 3.42 | 3.42 | 3.42 | 3.42 | 32.2 | 39.05 | 33.69 | 29.68 |
| Terrestrial Ecotoxicity | kg 1.4-DB eq | $1.22 \times 10^{-5}$ | $1.20 \times 10^{-5}$ | $1.17 \times 10^{-5}$ | $1.14 \times 10^{-5}$ | $5.26 \times 10^{-5}$ | $5.26 \times 10^{-5}$ | $5.26 \times 10^{-5}$ | $5.26 \times 10^{-5}$ | $2.95 \times 10^{-5}$ | $2.95 \times 10^{-5}$ | $2.95 \times 10^{-5}$ | $2.95 \times 10^{-5}$ | $1.39 \times 10^{-4}$ | $1.26 \times 10^{-4}$ | $1.19 \times 10^{-4}$ | $1.14 \times 10^{-4}$ |
| Photochemical Oxidation | kg $C_2H_4$ eq | $1.55 \times 10^{-6}$ | $1.51 \times 10^{-6}$ | $1.47 \times 10^{-6}$ | $1.43 \times 10^{-6}$ | $1.72 \times 10^{-6}$ | $1.72 \times 10^{-6}$ | $1.72 \times 10^{-6}$ | $1.72 \times 10^{-6}$ | $3.94 \times 10^{-6}$ | $3.94 \times 10^{-6}$ | $3.94 \times 10^{-6}$ | $3.94 \times 10^{-6}$ | $4.48 \times 10^{-5}$ | $4.49 \times 10^{-5}$ | $3.80 \times 10^{-5}$ | $3.29 \times 10^{-5}$ |

The manufacturing phase in the traditional system accounts for 90% of the "Global Warning Potential". Most of this impact (90%) is due, in particular, to the combustion of fossil fuels and the use of electricity during the firing process. The raw materials, extraction and transportation phases account for 3, 1, and 5%, respectively, for this impact category. The manufacturing phase is also the main cause of impact in all the other categories analyzed (between 51% and 86%). The manufacturing phase accounts for 92% of the 'Global Warning Potential' for bricks with 3% of olive pomace, reaching 93% and 94% for bricks containing 7% and 10% of this waste, respectively, as shown in Figure 6. The environmental profile obtained for the bricks containing olive pomace is very similar to that observed for the traditional bricks, where most of the impact categories are dominated by the manufacturing phase, between 86% and 49% (Figure 6).

Of all the stages evaluated, it is in the extraction stage where the environmental benefit produced by the addition of olive pomace to the mixture is the greatest, assuming, as shown in Figures 5 and 6, both in the traditional system and in the bricks with olive pomace, only 1% of the "Global Warning Potential". Extraction is also the most beneficial step in the "Photochemical Oxidation", "Ozone Depletion", "Acidification" and "Abiotic Depletion" categories, with values ranging from 1 to 3%. For the remaining categories ("Ecotoxicity", "Human Toxicity", and "Eutrophication"), the stage providing the greatest environmental benefit is that of raw materials, with values below 8%.

Figure 7 shows the impact categories analyzed associated with the traditional brick manufacturing process, to which a reference value of 100% has been assigned in order to establish comparisons with the rest of the materials tested. The results show that the increase in the amount of olive pomace added generates a positive environmental impact with respect to the traditional process in the categories "Depletion of Abiotic Resources", due to the lower consumption of raw materials due to the addition of waste, "Depletion of the Ozone Layer" due to the fact that the incorporation of waste reduces the extent of endothermic reactions that occur during the manufacture of the ceramic product, thus reducing fuel consumption, "Terrestrial Ecotoxicity", due to the impact savings achieved by avoiding the dumping of the olive pomace, thus avoiding the leaching of potentially toxic species, and "Photochemical Oxidation", due to the decrease in the impacts derived from atmospheric emissions produced during the firing process.

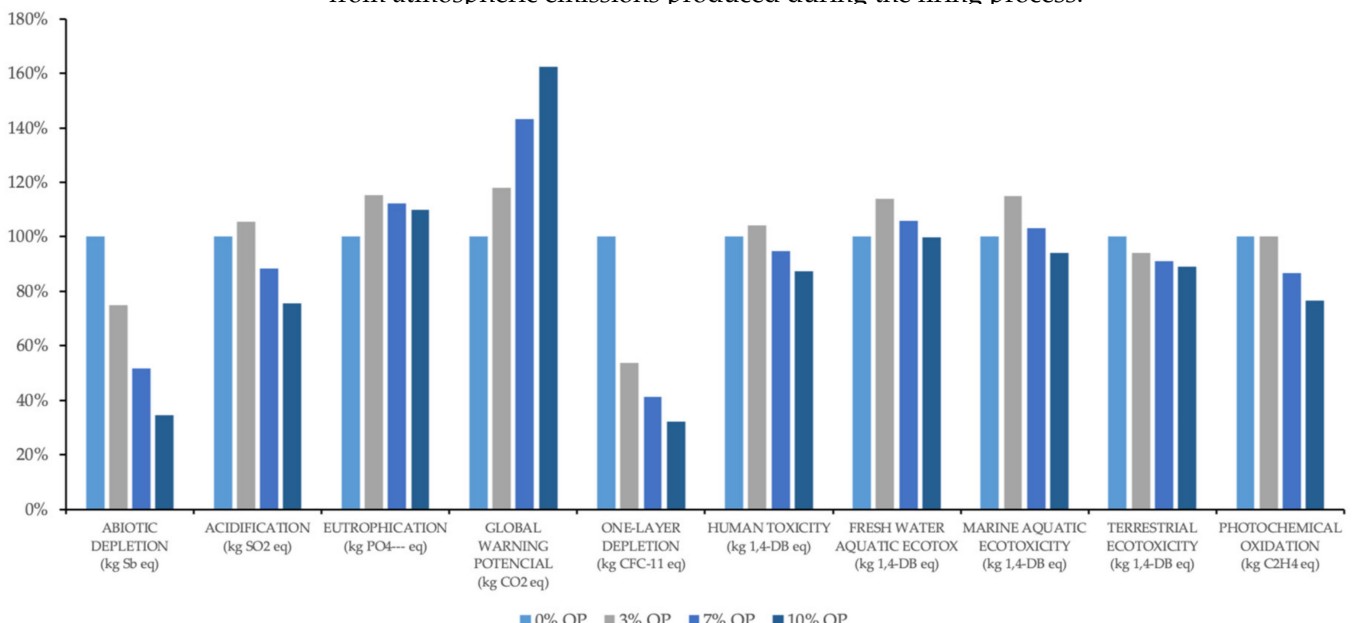

**Figure 7.** Impact categories analyzed using CML methodology.

On the other hand, the addition of olive pomace worsens the environmental performance of the bricks with respect to the traditional process only for the categories "Eutrophication" and "Freshwater Ecotoxicity". It is observed that "Seawater Ecotoxicity" also worsens, although with a slight benefit for bricks with 10% olive pomace.

Comparing bricks with different proportions of added olive pomace, it can be seen that in all environmental categories, except "Global Warming Potential", there is a decrease in the environmental impact caused by increasing the amount of waste added.

It should be noted that the analysis performed with both methodologies for the impact category "Global Warming Potential" (GWP) provides results with a high degree of accuracy, practically 99.5%.

Normalization is the calculation of the magnitude of a category indicator with respect to a reference information, i.e., it determines the relative magnitude of the LCIA results with respect to certain reference information in order to be able to interpret the LCA results in a simpler way. The criteria used to carry out the normalization are implicit in the methodologies that perform this step for impact assessment. Each impact assessment methodology uses its own reference system.

Figure 8 and Table 4 show the normalized results obtained by comparing the four scenarios investigated in this work. It can be seen that the impact category "Marine Aquatic Ecotoxicity" significantly exceeds the rest of the impact categories, a situation similar to that obtained by other authors [29–33]. These results may be due to the fact that the values provided by the normalization could be biased, either because the "Marine Aquatic Ecotoxicity" is too high or because the remaining impact categories are too low, given that the production of bricks with the addition of olive pomace is not considered an activity with high polluting repercussions for the marine environment.

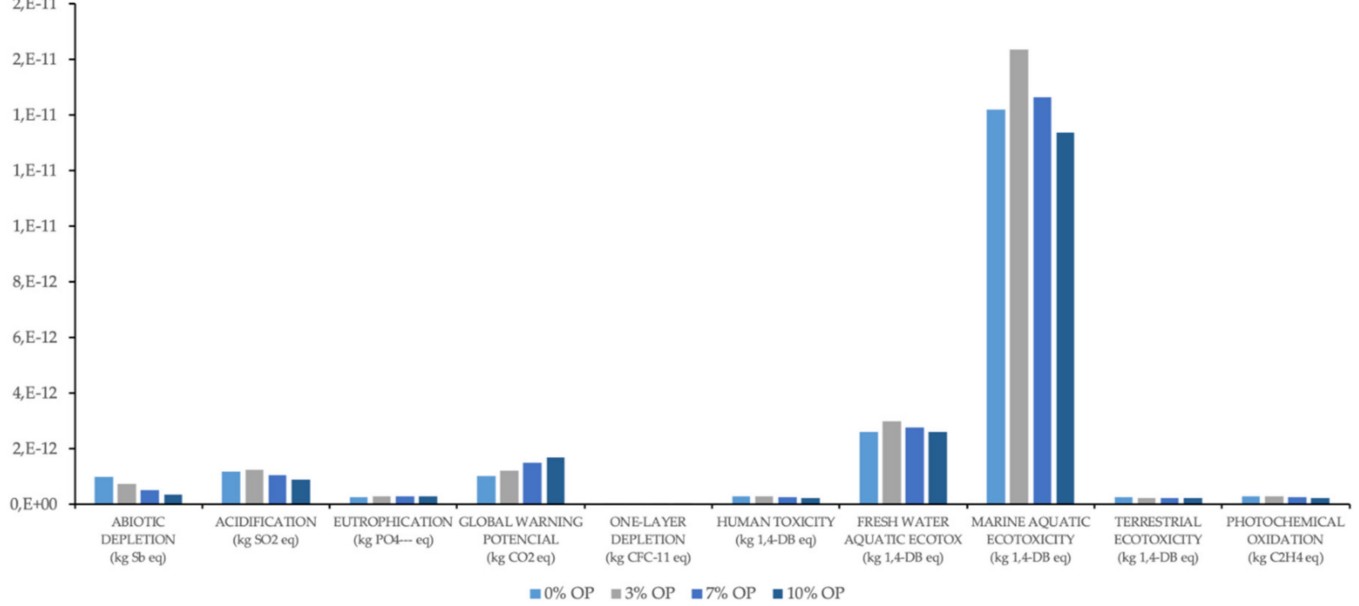

**Figure 8.** Impact categories analyzed using normalized CML.

**Table 4.** Normalized impacts associated with 1 kg of bricks incorporating 0, 3, 7, and 10 wt% OP.

| Impact Categories | Units | Raw Materials | | | | Extraction | | | | Transport | | | | Manufacturing | | | |
|---|---|---|---|---|---|---|---|---|---|---|---|---|---|---|---|---|---|
| | | 0% | 3% | 7% | 10% | 0% | 3% | 7% | 10% | 0% | 3% | 7% | 10% | 0% | 3% | 7% | 10% |
| Abiotic Depletion | kg Sb eq | $3.04 \times 10^{-14}$ | $2.97 \times 10^{-14}$ | $2.87 \times 10^{-14}$ | $2.79 \times 10^{-14}$ | $1.43 \times 10^{-14}$ | $1.43 \times 10^{-14}$ | $1.43 \times 10^{-14}$ | $1.43 \times 10^{-14}$ | $1.32 \times 10^{-13}$ | $1.32 \times 10^{-13}$ | $1.32 \times 10^{-13}$ | $1.32 \times 10^{-13}$ | $8.14 \times 10^{-13}$ | $5.65 \times 10^{-13}$ | $3.38 \times 10^{-13}$ | $1.69 \times 10^{-13}$ |
| Acidification | kg SO$_2$ eq | $6.72 \times 10^{-14}$ | $6.56 \times 10^{-14}$ | $6.34 \times 10^{-14}$ | $6.17 \times 10^{-14}$ | $4.07 \times 10^{-14}$ | $4.07 \times 10^{-14}$ | $4.07 \times 10^{-14}$ | $4.07 \times 10^{-14}$ | $1.22 \times 10^{-13}$ | $1.22 \times 10^{-13}$ | $1.22 \times 10^{-13}$ | $1.22 \times 10^{-13}$ | $9.54 \times 10^{-13}$ | $1.02 \times 10^{-12}$ | $8.19 \times 10^{-13}$ | $6.69 \times 10^{-13}$ |
| Eutrophication | kg PO$_4^{3-}$ eq | $2.13 \times 10^{-14}$ | $2.08 \times 10^{-14}$ | $2.01 \times 10^{-14}$ | $1.96 \times 10^{-14}$ | $2.57 \times 10^{-14}$ | $2.57 \times 10^{-14}$ | $2.57 \times 10^{-14}$ | $2.57 \times 10^{-14}$ | $2.96 \times 10^{-14}$ | $2.96 \times 10^{-14}$ | $2.96 \times 10^{-14}$ | $2.96 \times 10^{-14}$ | $1.87 \times 10^{-13}$ | $2.28 \times 10^{-13}$ | $2.20 \times 10^{-13}$ | $2.15 \times 10^{-13}$ |
| Global Warning Potencial | kg CO$_2$ eq | $2.93 \times 10^{-14}$ | $2.86 \times 10^{-14}$ | $2.77 \times 10^{-14}$ | $2.70 \times 10^{-14}$ | $1.51 \times 10^{-14}$ | $1.51 \times 10^{-14}$ | $1.51 \times 10^{-14}$ | $1.51 \times 10^{-14}$ | $5.65 \times 10^{-14}$ | $5.65 \times 10^{-14}$ | $5.65 \times 10^{-14}$ | $5.65 \times 10^{-14}$ | $9.32 \times 10^{-13}$ | $1.12 \times 10^{-12}$ | $1.38 \times 10^{-12}$ | $1.58 \times 10^{-12}$ |
| One-Layer Depletion | kg CFC$^{-11}$ eq | $1.25 \times 10^{-15}$ | $1.21 \times 10^{-15}$ | $1.17 \times 10^{-15}$ | $1.13 \times 10^{-15}$ | $3.32 \times 10^{-16}$ | $3.32 \times 10^{-16}$ | $3.32 \times 10^{-16}$ | $3.32 \times 10^{-16}$ | $6.13 \times 10^{-15}$ | $6.13 \times 10^{-15}$ | $6.13 \times 10^{-15}$ | $6.13 \times 10^{-15}$ | $2.16 \times 10^{-14}$ | $8.07 \times 10^{-15}$ | $4.49 \times 10^{-15}$ | $1.82 \times 10^{-15}$ |
| Human Toxicity | kg 1.4-DB eq | $2.29 \times 10^{-14}$ | $2.23 \times 10^{-14}$ | $2.16 \times 10^{-14}$ | $2.10 \times 10^{-14}$ | $6.62 \times 10^{-14}$ | $6.62 \times 10^{-14}$ | $6.62 \times 10^{-14}$ | $6.62 \times 10^{-14}$ | $4.47 \times 10^{-14}$ | $4.47 \times 10^{-14}$ | $4.47 \times 10^{-14}$ | $4.47 \times 10^{-14}$ | $1.42 \times 10^{-13}$ | $1.54 \times 10^{-13}$ | $1.28 \times 10^{-13}$ | $1.09 \times 10^{-13}$ |
| Fresh Water Aquatic Ecotox | kg 1.4-DB eq | $1.26 \times 10^{-13}$ | $1.24 \times 10^{-13}$ | $1.21 \times 10^{-13}$ | $1.19 \times 10^{-13}$ | $6.23 \times 10^{-14}$ | $6.23 \times 10^{-13}$ | $6.23 \times 10^{-13}$ | $6.23 \times 10^{-13}$ | $2.25 \times 10^{-13}$ | $2.25 \times 10^{-13}$ | $2.25 \times 10^{-13}$ | $2.25 \times 10^{-13}$ | $1.64 \times 10^{-12}$ | $2.01 \times 10^{-12}$ | $1.80 \times 10^{-12}$ | $1.65 \times 10^{-12}$ |
| Marine Aquatic Ecotoxicity | kg 1.4-DB eq | $5.27 \times 10^{-13}$ | $5.18 \times 10^{-13}$ | $5.06 \times 10^{-13}$ | $4.97 \times 10^{-13}$ | $2.50 \times 10^{-12}$ | $2.50 \times 10^{-12}$ | $2.50 \times 10^{-12}$ | $2.50 \times 10^{-12}$ | $1.07 \times 10^{-12}$ | $1.07 \times 10^{-12}$ | $1.07 \times 10^{-12}$ | $1.07 \times 10^{-12}$ | $1.01 \times 10^{-11}$ | $1.23 \times 10^{-11}$ | $1.06 \times 10^{-11}$ | $9.32 \times 10^{-12}$ |
| Terrestrial Ecotoxicity | kg 1.4-DB eq | $1.33 \times 10^{-14}$ | $1.31 \times 10^{-14}$ | $1.27 \times 10^{-14}$ | $1.24 \times 10^{-14}$ | $5.73 \times 10^{-14}$ | $5.73 \times 10^{-14}$ | $5.73 \times 10^{-14}$ | $5.73 \times 10^{-14}$ | $3.22 \times 10^{-14}$ | $3.22 \times 10^{-14}$ | $3.22 \times 10^{-14}$ | $3.22 \times 10^{-14}$ | $1.52 \times 10^{-13}$ | $1.37 \times 10^{-13}$ | $1.30 \times 10^{-13}$ | $1.24 \times 10^{-13}$ |
| Photochemical Oxidation | kg C$_2$H$_4$ eq | $8.49 \times 10^{-15}$ | $8.31 \times 10^{-15}$ | $8.05 \times 10^{-15}$ | $7.86 \times 10^{-15}$ | $9.43 \times 10^{-15}$ | $9.43 \times 10^{-15}$ | $9.43 \times 10^{-15}$ | $9.43 \times 10^{-15}$ | $2.16 \times 10^{-14}$ | $2.16 \times 10^{-14}$ | $2.16 \times 10^{-14}$ | $2.16 \times 10^{-14}$ | $2.46 \times 10^{-13}$ | $2.47 \times 10^{-13}$ | $2.09 \times 10^{-13}$ | $1.80 \times 10^{-13}$ |

## 4. Conclusions

To prevent the production of materials from affecting natural resources, it is necessary to promote the use of the best available techniques, as well as innovation in production plants to replace, as far as possible, the use of finite natural resources with the waste generated in the different production processes, closing product cycles. This implies a firm commitment to reuse and recycling, and always minimizing the transport of raw materials and products, promoting the use of available resources in local areas.

The ceramics industry generates impacts throughout its entire production cycle, from the extraction of the necessary resources to the final distribution of the product to the customer and the disposal of the waste generated. The LCA conducted in this research examines the brick production system from cradle to gate by comparing bricks produced in the traditional way with bricks to which olive pomace (alperujo) has been added at 3, 7, and 10% by weight, measuring its impact using IPCC and CML methodologies for a reference time of 100 years.

Regarding the potential environmental benefits of the incorporation of olive pomace into fired bricks, it has been observed that the stages of the life cycle that benefit the most are the "raw material" and "extraction" stages. Extraction accounts for only 1% of the "Global Warning Potential" in both the traditional system and in bricks with olive pomace, and between 1 and 3% for the environmental categories of "Photochemical Oxidation", "Ozone Depletion", "Acidification", and "Abiotic Depletion". The raw materials stage is the most environmentally beneficial in the "Ecotoxicity", "Human Toxicity", and "Eutrophication" categories, with values below 8%, mainly due to a reduced need for feedstock, which is covered by the waste added.

The least benefited stage would be the manufacturing, mainly due to the atmospheric emissions produced by the thermal decomposition of the raw materials used and the energy intensity required for the process, which, although reduced in some categories due to the exothermic decomposition of the olive pomace, is still high. The manufacturing phase in the traditional system accounts for 90% of the "Global Warning Potential", 92% of the "Global Warning Potential" for bricks with 3% alpeorujo, and 93% and 94% for bricks with 7 and 10 wt% olive pomace, respectively. The environmental profile for the other categories is similar, with no significant variations due to the olive pomace addition.

On the other hand, transport is not modified with the incorporation of the residue compared to the traditional system, as the emissions from the transport of the olive pomace have been assimilated as the corresponding part of the raw material it replaces.

The aggregate analysis carried out shows that the benefits of incorporating olive pomace into bricks are very limited. Although olive pomace bricks performed better in the categories of Abiotic Resource Depletion, Ozone Depletion, and Terrestrial Ecotoxicity, this was offset by higher impacts in the categories of Acidification, Eutrophication, Human Toxicity, Freshwater and Seawater Ecotoxicity, and Photochemical Oxidation. Finally, the higher amount of added olive pomace results in a slight decrease in environmental impact in all environmental categories except "Global Warming Potential".

In short, construction is one of the least environmentally sustainable sectors, generating high environmental costs, mainly due to the high consumption of resources and the large amount of waste produced. The brick manufacturing process has a negative environmental impact and is one of the most used materials on a daily basis in the construction sector worldwide. For this reason, the brick industry should implement technologies that consume less energy or use other more renewable energy sources. In this sense, an essential aspect is the use of alternative raw materials that can replace clay, such as waste, thus meeting the required technological properties while significantly reducing the environmental impact.

**Author Contributions:** Conceptualization, A.B.L.-G. and T.C.-P.; methodology, A.B.L.-G. and M.U.-R.; software, M.U.-R.; validation, A.B.L.-G. and M.U.-R.; formal analysis, A.B.L.-G.; investigation, A.B.L.-G. and M.U.-R.; resources, C.J.C.-C.; data curation, M.U.-R.; writing—original draft preparation, A.B.L.-G.; writing—review and editing, J.M.M.-M.; visualization, N.M.F.A.; supervision, T.C.-P.; project administration, C.M.-G.; funding acquisition, C.M.-G. All authors have read and agreed to the published version of the manuscript.

**Funding:** This research was conducted as a part of the EcoMetal Project (PID2019-109520RB-I00), "Can industrial and mining metalliferous wastes produce green lightweight aggregates? Applying the Circular Economy", funded by the Spanish Ministry of Economy and Competitiveness and FEDER (MINECO-FEDER).

**Institutional Review Board Statement:** Not applicable.

**Informed Consent Statement:** Not applicable.

**Data Availability Statement:** Not applicable.

**Acknowledgments:** The authors gratefully acknowledge this support. The authors also gratefully acknowledge the technical and human support provided by CICT of the University of Jaén and the University of Málaga (UJA, MINECO, Junta de Andalucía, FEDER).

**Conflicts of Interest:** The authors declare no conflict of interest.

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
