# Peer review of "Application of Life Cycle Assessment in the Environmental Study of Sustainable Ceramic Bricks Made with ‘alperujo’ (Olive Pomace)"

_applsci, doi:10.3390/app11052278_

Round 1
Reviewer 1 Report
The article is up to date with the topics connected with sustainability. Using olive pomace in bricks production is promising application and possible way to decrease carbon footprint.
The introduction is well organised however the literature survey is limited to the articles published before 2014 which might lead to lack of the newest knowledge in this scientific field.
The data is collected sufficiently and with accordance to such articles, however presenting results in table 1 which cover more than 2 pages is hard to read. In reviewers opinion division of this table for table according raw materials, separately extraction and transport, separately Energy and material inputs at bricks manufacturing plant etc. might be beneficial for the understanding where in particular phase of this cycle the influence at the environment is the highest. Also it is easier to analyse shorter tables.
Another remark is to the figure 7. Its probably in Spanish. It should be translated to English.
After revising this remarks manuscript can be published.
Author Response
We would like to thank the reviewer for taking the time to read our work and provide a number of recommendations, which have helped to improve our work. We will now proceed to respond to the reviewer’s comments.
The article is up to date with the topics connected with sustainability. Using olive pomace in bricks production is promising application and possible way to decrease carbon footprint.
The introduction is well organised however the literature survey is limited to the articles published before 2014 which might lead to lack of the newest knowledge in this scientific field.
The number of newest literature references has been expanded.
The data is collected sufficiently and with accordance to such articles, however presenting results in table 1 which cover more than 2 pages is hard to read. In reviewers opinion division of this table for table according raw materials, separately extraction and transport, separately Energy and material inputs at bricks manufacturing plant etc. might be beneficial for the understanding where in particular phase of this cycle the influence at the environment is the highest. Also it is easier to analyse shorter tables.
The table has been divided to improve the reader's understanding.
Another remark is to the figure 7. Its probably in Spanish. It should be translated to English.
The errata have been suitably corrected. We regret that they were overlooked when the manuscript was submitted.
After revising this remarks manuscript can be published.
All changes are indicated in red on the original manuscript. Thank you very much for your time and your words.

Reviewer 2 Report
There are some weaknesses through the manuscript which need improvement. Therefore, the submitted manuscript cannot be accepted for publication in this form, but it has a chance of acceptance after a revision. My comments and suggestions are as follows:
1- Abstract gives information on the main feature of the performed study, but some details about the conducted comparative LCA must be added. However, a concise abstract is needed.
2- The sentences in abstract are too long. For example, the first sentence is presented in five line. Rewritten of the abstract is necessary.
3- Authors must clarify necessity of the performed research.
4- Introduction is too short and the literature study must be enriched. As the manuscript deals with sensor for health monitoring, it is necessary to read and cite the relevant published papers: (a) https://doi.org/10.1016/j.apmt.2020.100689 (b) https://doi.org/10.1016/j.jclepro.2020.123259
5- It is necessary to mention the main references of standards. Geometries of specimens must be added to Fig. 1 and 2.
6- In 2.1 “Figures x-x show images …” and in 2.2 “there represented in Figure X, where …“. It seems the manuscript prepared without care. Text must be double check.
7- Life cycle flow chart illustrated in Fig. 3 must be explained in detail.
8- In its language layer, the manuscript should be considered for a professional proofreading. There are sentences which have to be rewritten.
9- The conclusion must be more than just a summary of the manuscript. Please provide all changes and reference update based on the proposed papers, by red color in the revised version.
Author Response
We would like to thank the reviewer for taking the time to read our work and provide a number of recommendations, which have helped to improve our work. We will now proceed to respond to the reviewer’s comments.
There are some weaknesses through the manuscript which need improvement. Therefore, the submitted manuscript cannot be accepted for publication in this form, but it has a chance of acceptance after a revision. My comments and suggestions are as follows:
1- Abstract gives information on the main feature of the performed study, but some details about the conducted comparative LCA must be added. However, a concise abstract is needed.
More information on the LCA study has been added.
2- The sentences in abstract are too long. For example, the first sentence is presented in five line. Rewritten of the abstract is necessary.
Some sentences have been restructured to improve the reader's understanding.
3- Authors must clarify necessity of the performed research.
This information has been incorporated into the revised paper.
4- Introduction is too short and the literature study must be enriched. As the manuscript deals with sensor for health monitoring, it is necessary to read and cite the relevant published papers: (a) https://doi.org/10.1016/j.apmt.2020.100689 (b) https://doi.org/10.1016/j.jclepro.2020.123259
The number of literature references has been expanded and recommended references have been added.
5- It is necessary to mention the main references of standards. Geometries of specimens must be added to Fig. 1 and 2.
This information has been incorporated into the revised paper.
6- In 2.1 “Figures x-x show images …” and in 2.2 “there represented in Figure X, where …“. It seems the manuscript prepared without care. Text must be double check.
The errata have been suitably corrected. We regret that they were overlooked when the manuscript was submitted.
7- Life cycle flow chart illustrated in Fig. 3 must be explained in detail.
This information has been incorporated into the revised paper.
8- In its language layer, the manuscript should be considered for a professional proofreading. There are sentences which have to be rewritten.
The article has been revised to improve English.
9- The conclusion must be more than just a summary of the manuscript.
This information has been incorporated into the revised paper.
Please provide all changes and reference update based on the proposed papers, by red color in the revised version.
All changes are indicated in red on the original manuscript. Thank you very much for your time and your words.

Round 2
Reviewer 1 Report
Paper can be accepted
Author Response
Thank you very much for your comments
Reviewer 2 Report
The paper has been improved and corresponding modifications have been conducted. In my opinion, the current version can be considered for publication.
Author Response
Thank you very much for your comments